# GENERATIVE HIERARCHICAL MODELS FOR PARTS, OBJECTS, AND SCENES

## ABSTRACT

Compositional structures between parts and objects are inherent in natural scenes. Modeling such compositional hierarchies via unsupervised learning can bring various benefits such as interpretability and transferability, which are important in many downstream tasks. In this paper, we propose the first deep latent variable model, called RICH, for learning Representation of Interpretable Compositional Hierarchies. At the core of RICH is a latent scene graph representation that organizes the entities of a scene into a tree structure according to their compositional relationships. During inference, taking top-down approach, RICH is able to use higher-level representation to guide lower-level decomposition. This avoids the difficult problem of routing between parts and objects that is faced by bottom-up approaches. In experiments on images containing multiple objects with different part compositions, we demonstrate that RICH is able to learn the latent compositional hierarchy and generate imaginary scenes.

## 1 INTRODUCTION

Compositional hierarchies prevail in natural scenes where primitive entities are recursively composed into more abstract entities. Modeling such compositional generative process allows discovery of modular primitives that can be reused across a variety of scenes. Hence, it would bring interpretability and transferability, in which current deep learning models are not quite successful. Due to expensive labeling, such compositional relationships should ideally be learned in an unsupervised manner. Unsupervised approaches can also provide more flexibility and generalization ability since the model is allowed to choose the most appropriate compositional hierarchy for a given scenario.

Despite its importance, there has not been much work on unsupervised generative modeling of the compositional hierarchy. Earlier work on hierarchical representation learning (Lee et al., 2009) obtains a feature hierarchy that captures concepts at different levels of abstraction, with no explicit modeling of composition. Recent researches on deep latent variable models (Maaløe et al., 2019; Zhao et al., 2017; Sønderby et al., 2016; Bachman, 2016) mainly focus on architectural designs and training methods that harness the full expressive power of hierarchical generative models. Although they have shown impressive generation quality and disentanglement of learned representation, the compositional hierarchy is still not captured in a modular and interpretable way. To obtain interpretable scene representation, recent work (Tieleman, 2014; Eslami et al., 2016; Crawford & Pineau, 2019; Wu et al., 2017; Yao et al., 2018; Romaszko et al., 2017; Deng et al., 2019) has introduced domain-specific decoders that take object pose and appearance information as input and render the object in a similar way to graphics engines. This forces the encoder to invert the rendering process, producing interpretable object-wise pose and appearance representation.

In this paper, we extend the interpretable object-wise representation to the hierarchical setting. We propose a deep generative model, called RICH (Representation of Interpretable Compositional Hierarchies), that can use its hierarchy to represent the compositional relationships among interpretable symbolic entities like parts, objects, and scenes. To this end, taking inspiration from capsule networks (Sabour et al., 2017; Hinton et al., 2018) and the rendering process of computer graphics, we propose a *probabilistic scene graph representation* that describes the compositional hierarchy as a latent tree. The nodes in the tree correspond to entities in the scene, while the edges indicate the compositional relationships among these entities. We associate an appearance latent with each node to summarize all lower-level composition, and a pose latent with each edge to specify the

transformation from the current level to the upper level. To enforce interpretability, the probabilistic scene graph is then paired with a decoder that renders the scene graph by recursively applying the specified transformations. We also introduce learnable templates for the primitive entities. Once learned, RICH is able to generate all lower-level latents and render a partial scene given the latent at a specific level.

To infer the scene graph is, however, challenging, since both the tree structure and the latent variables need to be simultaneously inferred. Capsule networks have provided a bottom-up solution to learning the tree structure, but it faces the difficult routing problem caused by the exponentially many possible compositions. Instead, RICH takes a top-down approach that avoids the routing problem. The intuition is that for a given scene, it is natural to first decompose it into high-level objects. If we devote our attention to one of the objects, we can then figure out its constituent parts. In cases where parts are close or have occlusion, we expect the appearance latent of the higher-level object to guide lower-level decomposition, since it summarizes the typical composition for that object.

The contributions of this paper are as follows. We propose RICH, the first interpretable representation learning model for compositional hierarchies through probabilistic latent variable modeling. We then implement a three-level prototype of RICH and demonstrate its effectiveness in extensive experiments. RICH is able to learn the hierarchical scene graph representation from images containing multiple compositional objects. Further, it shows decent generation quality and generalization ability to unseen number of objects.

## 2 RELATED WORK

**Interpretable object-wise representation.** AIR (Eslami et al., 2016) is the first generative model that learns interpretable object-wise scene representation. It is able to assign a latent vector ($z^{\text{pres}}, \mathbf{z}^{\text{where}}, \mathbf{z}^{\text{what}}$) to each object in the scene, describing the presence, size, center position, and appearance of the object. SPAIR (Crawford & Pineau, 2019) improves the scalability of AIR to images containing a large number of objects. It divides the image into spatially distributed cells, and auto-regressively infers the latent vector for each cell. This crucially reduces the search space for individual cells, since they are each responsible only for explaining objects near themselves. RICH builds upon SPAIR to infer the structure of the probabilistic scene graph. To enable efficient hierarchical inference, we use mean-field approximation for the posterior, allowing inference of all cells to be done in parallel.

**Hierarchical scene representation.** Modeling the part-whole relationship in scenes has attracted growing interest, and it has been utilized for improving image classification, parsing, and segmentation. Two representative models that have inspired our work are hierarchical compositional models (HCMs) and capsule networks. In HCMs (Zhu et al., 2008), the hierarchical structure is represented as a graph, where leaf nodes interact with image segments, and upper-level nodes store the average position and orientation of lower-level nodes (with respect to the image coordinates). In capsule networks (Sabour et al., 2017; Hinton et al., 2018), the part-whole relationship is used for achieving viewpoint invariance. The key insight is that the relative pose of parts with respect to objects is viewpoint invariant, and is thus suitable to be learned as network weights. However, neither of these two approaches uses generative modeling, and they have been applied only to scenes with one dominant object. More recently, the part-whole relationship has been explored in modeling 3D shapes (Tulsiani et al., 2017; Li et al., 2017; Zhu et al., 2018) and motion decomposition (Xu et al., 2019). Although they have employed generative modeling, the latent space is not hierarchically structured, and typically only contains the top-level representation. In contrast, RICH represents the entire scene graph in latent space, modeling both the hierarchical structure and the object pose and appearance at each level as latent variables. This representation is arguably richer, and enables simultaneous learning of a collection of generative models for parts, objects, and scenes.

## 3 THE PROPOSED MODEL: RICH

RICH (Representation of Interpretable Compositional Hierarchies) is a generative model that captures the recursive compositional structure inherent in natural scenes. It builds a tree-structured representation similar to scene graphs in computer graphics (Foley et al., 1996). The nodes in the tree describe entities at various levels of abstraction in the scene, and the edges indicate the compo-

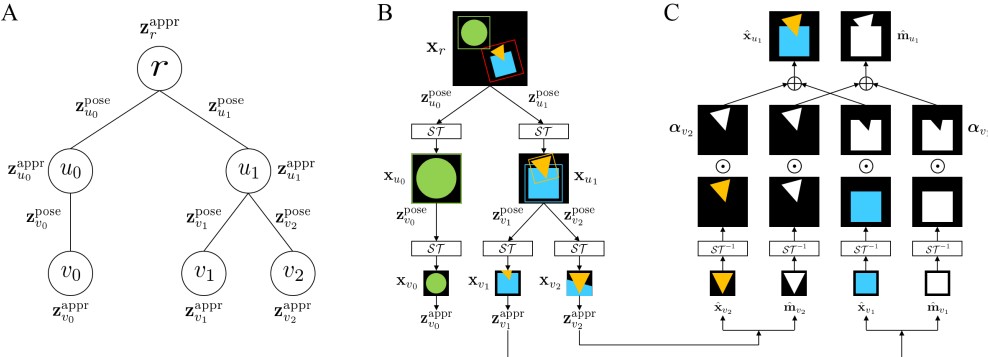

**Figure 1:** *(A)* Probabilistic scene graph representation. Each node represents an entity in the scene, and is associated with an appearance latent. Each edge is associated with a relative pose latent that specifies the coordinate transformation between the child node and the parent node. *(B)* Top-down inference process. Inference combines information from glimpse regions and higher-level appearance latents (not shown here). Bounding boxes indicate inferred pose latents. *(C)* Recursive decoding process (a single recursive step shown). The image patch $\hat{\mathbf{x}}_{u_1}$ and mask $\hat{\mathbf{m}}_{u_1}$ of an internal node $u_1$ are decoded from the image patches and masks of all its children nodes.

sitional relationships among these entities. Specifically, each leaf node represents a primitive entity that is not further decomposed. Each internal node represents an abstract entity that is composed from its children nodes. The composition is specified by the relative pose of each child node with respect to the parent node, and this pose information is stored on the corresponding edges.

## 3.1 GENERATIVE PROCESS

To make a generative model, we associate an appearance vector $\mathbf{z}_v^{\text{appr}}$ with each node $v$, and a pose vector $\mathbf{z}_v^{\text{pose}}$ with the edge between node $v$ and its parent $pa(v)$, as shown in Figure 1A. The intuition is that $\mathbf{z}_{pa(v)}^{\text{appr}}$ represents the entity at $pa(v)$ in its canonical pose, summarizing all lower-level composition in the subtree rooted at $pa(v)$. Conditioning on $\mathbf{z}_{pa(v)}^{\text{appr}}$, we can then sample $\mathbf{z}_v^{\text{appr}}$ and $\mathbf{z}_v^{\text{pose}}$, the relative pose of $v$ with respect to $pa(v)$. Thus, the latent vectors in any subtree can be recursively generated. Let $V$ be the set of all nodes, $r \in V$ be the root node, and $L \subseteq V$ be the set of leaf nodes. The generative model for the entire scene $\mathbf{x}$ can be written as follows:

$$p(\mathbf{x}) = \int p(\mathbf{x} \mid \mathbf{z}_{V\setminus\{r\}}^{\text{pose}}, \mathbf{z}_L^{\text{appr}}) \, p(\mathbf{z}_r^{\text{appr}}) \prod_{v \in V\setminus\{r\}} p(\mathbf{z}_v^{\text{pose}} \mid \mathbf{z}_{pa(v)}^{\text{appr}}) \, p(\mathbf{z}_v^{\text{appr}} \mid \mathbf{z}_{pa(v)}^{\text{appr}}) \, d\mathbf{z}, \qquad (1)$$

where we assume conditional independence among all latents $\mathbf{z}_v^{\text{pose}}$ and $\mathbf{z}_v^{\text{appr}}$ that have the same $pa(v)$. This gives disentangled and interpretable scene representation.

We design the decoder $p(\mathbf{x}|\mathbf{z}_{V\setminus\{r\}}^{\text{pose}}, \mathbf{z}_L^{\text{appr}})$ to closely follow the rendering process from a given scene graph. First, for each leaf node $v \in L$, we use a neural network to decode its appearance vector into a small image patch $\hat{\mathbf{x}}_v$ and a (close to) binary mask $\hat{\mathbf{m}}_v$ the same size as $\hat{\mathbf{x}}_v$. Here we assume that $\hat{\mathbf{x}}_v$ has already been masked by $\hat{\mathbf{m}}_v$, meaning $\hat{\mathbf{x}}_v(i, j) = 0$ for all pixel locations $(i, j)$ where $\hat{\mathbf{m}}_v(i, j) = 0$. We then recursively compose these primitive patches into the entire scene by applying affine transformations level by level. Specifically, let $u$ be an internal node, and $ch(u)$ be the set of its children. We compose the higher-level image patch $\hat{\mathbf{x}}_u$ and mask $\hat{\mathbf{m}}_u$ as follows:

$$\hat{\mathbf{x}}_u = \sum_{v \in ch(u)} \boldsymbol{\alpha}_v \odot \mathcal{ST}^{-1}(\hat{\mathbf{x}}_v, \mathbf{z}_v^{\text{pose}}), \quad \hat{\mathbf{m}}_u = \sum_{v \in ch(u)} \boldsymbol{\alpha}_v \odot \mathcal{ST}^{-1}(\hat{\mathbf{m}}_v, \mathbf{z}_v^{\text{pose}}), \qquad (2)$$

where a spatial transformer $\mathcal{ST}$ (Jaderberg et al., 2015) is used to properly place $\hat{\mathbf{x}}_v$ and $\hat{\mathbf{m}}_v$ into the coordinate frame of the parent node $u$, according to the scaling, rotation and translation parameters given by $\mathbf{z}_v^{\text{pose}}$. In addition, $\mathbf{z}_v^{\text{pose}}$ also provides relative depth information that helps deal with occlusion. Entities with smaller depth will appear in front of entities with larger depth. This is enforced by a transparency map $\boldsymbol{\alpha}_v$ that assigns pixel-wise weights to each transformed patch according to its relative depth. See Figure 1C for an illustration. To ensure that unoccluded part of entities will

remain visible, we normalize $\boldsymbol{\alpha}_v$ after applying the transformed mask $\tilde{\mathbf{m}}_v = \mathcal{ST}^{-1}(\hat{\mathbf{m}}_v, \mathbf{z}_v^{\text{pose}})$, namely for all pixel locations $(i, j)$,

$$\boldsymbol{\alpha}_v(i, j) = 0 \text{ if } \tilde{\mathbf{m}}_v(i, j) = 0, \quad \sum_{v \in ch(u)} \boldsymbol{\alpha}_v(i, j) = 1 \text{ if } \sum_{v \in ch(u)} \tilde{\mathbf{m}}_v(i, j) > 0. \quad (3)$$

The final decoder output $p(\mathbf{x} | \mathbf{z}_{V\setminus\{r\}}^{\text{pose}}, \mathbf{z}_L^{\text{appr}}) = \mathcal{N}(\hat{\mathbf{x}}_r, \sigma^2 \mathbf{I})$, where $\sigma$ is a hyperparameter.

## 3.2 INFERENCE AND LEARNING

Since computing $p(\mathbf{x})$ in Equation 1 is intractable, we train RICH with variational inference. The approximate posterior is designed to factorize $\mathbf{z}$ in the top-down fashion similar to the generative process, such that a higher-level appearance representation guides lower-level decomposition:

$$p(\mathbf{z} \mid \mathbf{x}) \approx q(\mathbf{z} \mid \mathbf{x}) = q(\mathbf{z}_r^{\text{appr}} \mid \mathbf{x}) \prod_{v \in V\setminus\{r\}} q(\mathbf{z}_v^{\text{pose}} \mid \mathbf{z}_{pa(v)}^{\text{appr}}, \mathbf{x}_{pa(v)}) \, q(\mathbf{z}_v^{\text{appr}} \mid \mathbf{z}_{pa(v)}^{\text{appr}}, \mathbf{x}_v). \quad (4)$$

Here $\mathbf{x}_v$ is the region of input image $\mathbf{x}$ that corresponds to the entity that node $v$ represents. This region is specified by all the predicted pose vectors along the path from root $r$ to node $v$. More precisely, we define $\mathbf{x}_r = \mathbf{x}$, and recursively extract $\mathbf{x}_v = \mathcal{ST}(\mathbf{x}_{pa(v)}, \mathbf{z}_v^{\text{pose}})$, as shown in Figure 1B. Notice that the relative pose $\mathbf{z}_v^{\text{pose}}$ of $v$ with respect to $pa(v)$ needs to be inferred from $\mathbf{x}_{pa(v)}$. After applying spatial transformer $\mathcal{ST}$, we assume the captured $\mathbf{x}_v$ is in its canonical pose. This top-down inference process avoids the challenging routing problem in capsule networks (Sabour et al., 2017; Hinton et al., 2018), leading to more efficient inference. In cases where entities are close or have occlusion, the higher-level appearance vector $\mathbf{z}_{pa(v)}^{\text{appr}}$ could provide some guidance on separating these entities.

In general, all latents are assumed to be continuous, with both prior and posterior being Gaussian distributions. However, it may bring additional flexibility and interpretability to introduce some discrete latents, as we explain in Section 3.3. For continuous latents, we compute posterior via precision-weighted combination similar to Ladder-VAE (Sønderby et al., 2016), and use reparameterization trick (Kingma & Welling, 2013) to sample from the posterior. For discrete latents, we use Gumbel-Softmax trick (Jang et al., 2016; Maddison et al., 2016). Thus, the entire model can be trained end-to-end via backpropagation to maximize the following evidence lower bound (ELBO):

$$\mathcal{L} = \mathbb{E}_{q(\mathbf{z}|\mathbf{x})}[\log p(\mathbf{x} \mid \mathbf{z}_{V\setminus\{r\}}^{\text{pose}}, \mathbf{z}_L^{\text{appr}})] - D_{\text{KL}}[q(\mathbf{z}_r^{\text{appr}} \mid \mathbf{x}) \parallel p(\mathbf{z}_r^{\text{appr}})] \quad (5)$$

$$- \sum_{v \in V\setminus\{r\}} \mathbb{E}_{q(\mathbf{z}|\mathbf{x})}[D_{\text{KL}}[q(\mathbf{z}_v^{\text{pose}} \mid \mathbf{z}_{pa(v)}^{\text{appr}}, \mathbf{x}_{pa(v)}) \parallel p(\mathbf{z}_v^{\text{pose}} \mid \mathbf{z}_{pa(v)}^{\text{appr}})]]$$

$$- \sum_{v \in V\setminus\{r\}} \mathbb{E}_{q(\mathbf{z}|\mathbf{x})}[D_{\text{KL}}[q(\mathbf{z}_v^{\text{appr}} \mid \mathbf{z}_{pa(v)}^{\text{appr}}, \mathbf{x}_v) \parallel p(\mathbf{z}_v^{\text{appr}} \mid \mathbf{z}_{pa(v)}^{\text{appr}})]].$$

## 3.3 IMPLEMENTATION DETAILS

**Structural inference.** In our description above, we have assumed that the tree structure is already known. We now relax this assumption and introduce structural inference. First, we set a maximum out-degree for each node so that the number of all possible structures is bounded. For simplicity, in our implementation nodes within one level share the same maximum out-degree. To determine the structure, it then suffices to specify the presence of each possible edge. Hence, for an arbitrary edge between node $v$ and its parent, we introduce a Bernoulli variable $z_v^{\text{pres}}$ to indicate its presence. If $z_v^{\text{pres}} = 0$, meaning the edge is not present, then $\mathbf{z}_v^{\text{pose}}$ along with all latents in the subtree rooted at $v$ are excluded from the representation. To encourage sparse structures, we initialize the prior $p(z_v^{\text{pres}} \mid \mathbf{z}_{pa(v)}^{\text{appr}})$ and the posterior $q(z_v^{\text{pres}} \mid \mathbf{z}_{pa(v)}^{\text{appr}}, \mathbf{x}_{pa(v)})$ to have small Bernoulli parameters.

**Node grounding.** Due to the symmetric tree structure, there are numerous equivalent entity-to-node assignments for a given scene, each yielding a different permutation of the pose vectors. This can cause difficulties in the learning process. In particular, the model has to learn a consistent assignment strategy such that the pose vector at each edge can be well captured by a unimodal Gaussian distribution. To alleviate this problem, we impose some inductive bias on the assignment strategy. Inspired by SPAIR (Crawford & Pineau, 2019), for each internal node $u$, we divide $\mathbf{x}_u$ into a grid of $N_u$ cells, where $N_u$ is the maximum out-degree of $u$. Each child of $u$ is assigned to one

of these cells, and is responsible for explaining only the entity whose center position is within that cell. Assuming that $\mathbf{x}_u$ captures the entity $u$ in its canonical pose, this assignment strategy ensures that each child of node $u$ is almost always associated with the same component of entity $u$. To deal with occlusion, we let neighboring cells have slight overlap.

**Primitive templates.** It is often reasonable to assume that the vast number of complex entities can be composed from only a modest number of primitive entities. Identifying such primitive entities through discrete latent variables would bring additional interpretability. Hence, we introduce an external memory $\mathbf{M}$ with variational addressing (Bornschein et al., 2017) to learn a set of primitive templates. For each leaf node $v \in L$, we decompose its appearance vector into $\mathbf{z}_v^{\text{appr}} = (\mathbf{z}_v^{\text{addr}}, \mathbf{z}_v^{\text{what}})$. Here, $\mathbf{z}_v^{\text{addr}}$ is a one-hot vector that points one of the templates in $\mathbf{M}$, and $\mathbf{z}_v^{\text{what}}$ is a continuous vector that explains the remaining variability. We assume that $\mathbf{z}_{pa(v)}^{\text{appr}}$ captures the identity but not the exact appearance of entity $v$, since $\mathbf{z}_{pa(v)}^{\text{appr}}$ is intended for summarization. Therefore, we factorize the prior and the posterior as follows:

$$p(\mathbf{z}_v^{\text{appr}} \mid \mathbf{z}_{pa(v)}^{\text{appr}}) = p(\mathbf{z}_v^{\text{addr}} \mid \mathbf{z}_{pa(v)}^{\text{appr}}, \mathbf{M}) \, p(\mathbf{z}_v^{\text{what}} \mid \mathbf{M}[\mathbf{z}_v^{\text{addr}}]), \tag{6}$$

$$q(\mathbf{z}_v^{\text{appr}} \mid \mathbf{z}_{pa(v)}^{\text{appr}}, \mathbf{x}_v) = q(\mathbf{z}_v^{\text{addr}} \mid \mathbf{z}_{pa(v)}^{\text{appr}}, \mathbf{M}, \mathbf{x}_v) \, q(\mathbf{z}_v^{\text{what}} \mid \mathbf{M}[\mathbf{z}_v^{\text{addr}}], \mathbf{x}_v), \tag{7}$$

where $\mathbf{M}$ is considered as model parameter, and $\mathbf{M}[\mathbf{z}_v^{\text{addr}}]$ is the deterministically retrieved memory content. We implement $\mathbf{M}$ to be a stack of low-dimensional embeddings of templates. To decode from $\mathbf{z}_v^{\text{appr}}$, we first retrieve the embedding indexed by $\mathbf{z}_v^{\text{addr}}$, and decode it into a single-channel image patch. This serves as both the template and the mask, namely $\hat{\mathbf{m}}_v = g(\mathbf{M}[\mathbf{z}_v^{\text{addr}}])$. We then apply multiplicative modification controlled by $\mathbf{z}_v^{\text{what}}$ and obtain $\hat{\mathbf{x}}_v = \hat{\mathbf{m}}_v \odot h(\mathbf{z}_v^{\text{what}})$. Here, $\hat{\mathbf{x}}_v$ has the same number of channels as the input $\mathbf{x}$, and both $g(\cdot)$ and $h(\cdot)$ are implemented as spatial broadcast decoders (Watters et al., 2019).

**Canonical size.** To avoid undesired effects in cascaded affine transformations, we constrain the spatial transformer to always preserve aspect ratio. Thus, for a square input image, each entity is assumed to occupy a square image region. Considering the various possible compositions, this region may not capture the entity's canonical size well. Hence, we introduce a latent variable $z_u^{\text{ratio}}$ to represent the aspect ratio of entity $u$. This could give a tighter bounding box inside the square region. We consider $z_u^{\text{ratio}}$ as part of the appearance vector, namely $\mathbf{z}_u^{\text{appr}} = (\mathbf{z}_u^{\text{what}}, z_u^{\text{ratio}})$, and factorize the prior and the posterior as follows:

$$p(\mathbf{z}_u^{\text{appr}} \mid \mathbf{z}_{pa(u)}^{\text{appr}}) = p(\mathbf{z}_u^{\text{what}} \mid \mathbf{z}_{pa(u)}^{\text{appr}}) \, p(z_u^{\text{ratio}} \mid \mathbf{z}_u^{\text{what}}), \tag{8}$$

$$q(\mathbf{z}_u^{\text{appr}} \mid \mathbf{z}_{pa(u)}^{\text{appr}}, \mathbf{x}_u) = q(\mathbf{z}_u^{\text{what}} \mid \mathbf{z}_{pa(u)}^{\text{appr}}, \mathbf{x}_u) \, q(z_u^{\text{ratio}} \mid \mathbf{z}_u^{\text{what}}, \mathbf{x}_u). \tag{9}$$

For simplicity, we introduce $z_u^{\text{ratio}}$ only for intermediate nodes $u \in V \setminus (L \cup \{r\})$. To properly learn $z_u^{\text{ratio}}$, we feed it as an additional argument to the spatial transformer. This has two effects. First, the region outside the bounding box given by $z_u^{\text{ratio}}$ is masked out during transformation in both inference and reconstruction. This forces the bounding box to capture the entity in its entirety. Second, during training, we inject zero-mean Gaussian noise to the reconstruction inside the bounding box region. The noise level is annealed as training proceeds. This encourages tight bounding boxes without affecting generation quality.

# 4 EXPERIMENTS

## 4.1 DATASETS AND A THREE-LEVEL PROTOTYPE

We have implemented a prototype of RICH with part-, object-, and scene-level representation. We will refer to the latents as $\mathbf{z}_P$, $\mathbf{z}_O$, and $\mathbf{z}_S$ respectively. The maximum out-degree is set to be 4 for each internal node. For evaluation, we have made two datasets of 2D and 3D scenes. Both datasets contain $128 \times 128$ color images, split into 64000 for training, 12800 for validation, and 12800 for testing. They present challenges of (i) multi-pose, variable number of objects and parts, (ii) multiple occurrences of the same type of objects and parts within one scene, and (iii) severe occlusion in 3D scenes. In making each dataset, we first choose a set of primitive shapes to be the parts, and then construct the objects and scenes by recursively composing these parts. Specifically, we have chosen three shapes as parts, and defined ten types of objects in terms of the identity of the constituent parts

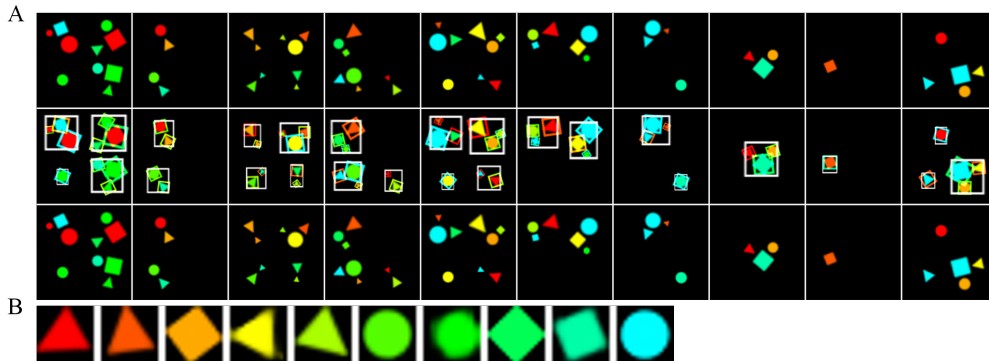

**Figure 2:** Qualitative results on 2D dataset. *(A) (Top)* Input image. *(Middle)* Input image superimposed with predicted bounding boxes, drawn according to $\mathbf{z}_O^{\mathrm{pres}}$, $\mathbf{z}_O^{\mathrm{pose}}$, $\mathbf{z}_O^{\mathrm{ratio}}$, $\mathbf{z}_P^{\mathrm{pres}}$, $\mathbf{z}_P^{\mathrm{pose}}$ and $\mathbf{z}_P^{\mathrm{addr}}$. *(Bottom)* Reconstruction. *(B)* Learned part-level templates. Template colors indicate identity. Part bounding box colors indicate the chosen template.

and their relative position, scale, and orientation. Among these ten types, three contain a single part, another three contain two parts, and the remaining four contain three parts. To construct a scene, we first randomly sample the number of objects (between one and four) and their types, and then instantiate these objects. This means for each object, we choose a random color for each of its parts, apply random scaling (within 10% of object size), and draw it at a random position in the scene. In 2D case, the instantiation process also includes random perturbation of parts and random rotation of the object as a whole. We ensure that different objects have minimal overlap. In 3D case, we use MuJoCo (Todorov et al., 2012) to place the objects on a plane, and then take observations from ten different viewpoints, some of which can lead to severe occlusion.

## 4.2 SCENE DECOMPOSITION

RICH is able to give interpretable, tree-structured decomposition of scenes into objects and parts. We visualize such decomposition in Figure 2 and Figure 3 for 2D and 3D scenes respectively, where we also show the learned memory templates for parts. Notice that the templates should be in gray scale, but for visualization purposes we have assigned a color to each template. The bounding boxes are drawn on top of the input images, according to the inferred pose of objects and parts with $z^{\mathrm{pres}} = 1$. Object bounding boxes are drawn in white, while part bounding boxes are drawn in color to indicate the template chosen for each part.

We find that the templates have learned the appearance of parts at several canonical poses (rotation in 2D and viewpoint in 3D), and RICH predicts the pose of parts with respect to these canonical poses. This makes the decomposition even more interpretable. Moreover, equipped with templates, RICH is able to correctly identify the parts even when they have severe occlusion. See Figure 3A third row where a ball occludes an equally sized cube. This example (and many others) also demonstrate that RICH can successfully deal with objects composed of multiple parts that are of the same type.

In addition to part-level templates, we believe that the learned object-level $\mathbf{z}_O^{\mathrm{appr}}$ also helps scene decomposition, especially when there is ambiguity in part assignment and occlusion between objects. For example, in Figure 2A first column, the triangle and circle near the center are close to each other and may well constitute an object. However, because this pose configuration is relatively rare in the training set (compare second column), RICH has correctly rejected this composition and instead assigned these two parts to separate objects, which better agrees with the training distribution. In Figure 3A fourth row, object 1 is occluded by object 3. RICH has successfully detected object 1 and added in the reconstruction a ball of the same color as the occluded part. This is quite reasonable since the augmented object is one of our predefined types and appears frequently in the dataset.

To quantify RICH's ability of scene decomposition and representation learning, we report absolute counting error, precision, and recall for detection of objects and parts in Table 1, and compare the negative log-likelihood of RICH with a VAE (Kingma & Welling, 2013) baseline in Table 2. Here the counting error measures the absolute difference between the predicted and true number of objects

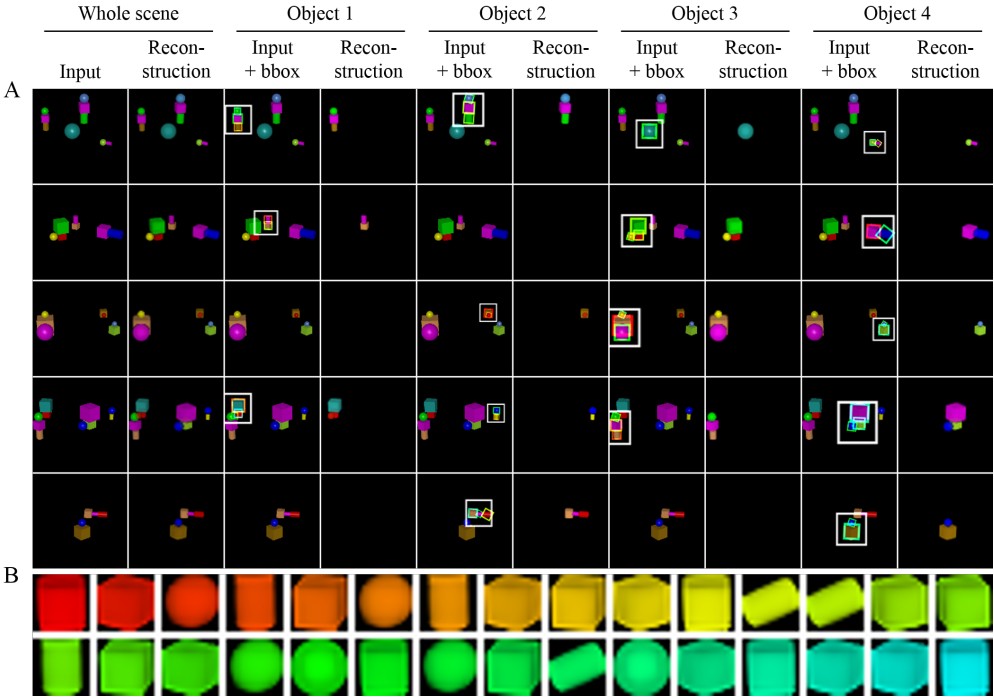

**Figure 3:** Qualitative results on 3D dataset. *(A)* Each row shows the overall reconstruction, and the predicted bounding boxes and reconstruction from each object cell for a given input image. *(B)* Learned part-level templates. Template colors indicate identity. Part bounding box colors indicate the chosen template.

**Table 1:** Quantitative results on object and part detection.

| Dataset | 2D dataset | | | 3D dataset | | |
|---|---|---|---|---|---|---|
| Training set | 1∼4 objects | 1&3 objects | | 1∼4 objects | 1&3 objects | |
| Test set | 1∼4 objects | 2 objects | 4 objects | 1∼4 objects | 2 objects | 4 objects |
| Object count error | 0.00083 | 0.0014 | 0.00036 | 0.094 | 0.26 | 0.47 |
| Object precision | 0.9985 | 0.9987 | 0.9996 | 0.9639 | 0.9157 | 0.9581 |
| Object recall | 0.9984 | 0.9982 | 0.9995 | 0.9597 | 0.9758 | 0.8462 |
| Part count error | 0.0086 | 0.011 | 0.014 | 0.80 | 1.1 | 1.3 |
| Part precision | 0.9991 | 0.9988 | 0.9991 | 0.8282 | 0.7579 | 0.8100 |
| Part recall | 0.9989 | 0.9985 | 0.9993 | 0.9116 | 0.9258 | 0.8347 |

and parts. To obtain precision and recall, we need to match the predictions with the groundtruth. We set the matching priority as the distance between the predicted and true center positions, namely closer pairs of prediction and groundtruth will be matched first. We only match the pair if their distance is less than 10 pixels (less than half of the size of large parts). This ensures that the matched predictions will have approximately correct center positions. The VAE baseline shares the same scene-level encoder with RICH, and uses sub-pixel convolution (Shi et al., 2016) for the decoder. We approximate the negative log-likelihood using 50 importance-weighted samples. The counting error, precision, and recall are also averaged over 50 samples from the posterior. As can be seen from Table 1, RICH gives almost perfect detection of objects and parts on 2D dataset, and still performs reasonably well on the challenging 3D dataset. We observe that RICH tends to split a long cylinder into two parts, leading to the drop in precision for parts.

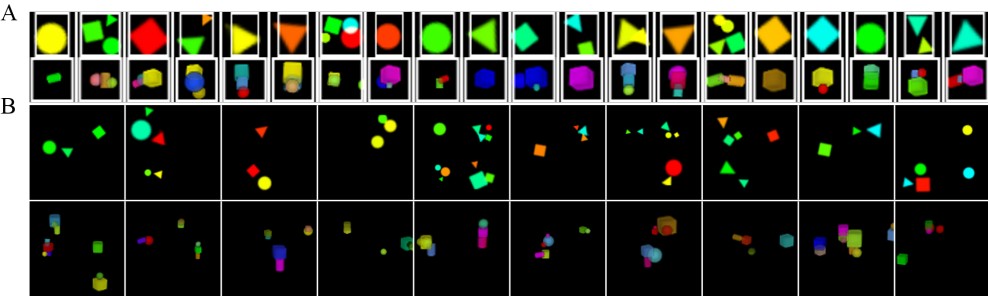

**Figure 4:** *(A)* Generated objects on *(Top)* 2D dataset and *(Bottom)* 3D dataset. White boxes indicate aspect ratio, and are drawn according to $\mathbf{z}_O^{\text{ratio}}$. *(B)* Generated scenes on *(Top)* 2D dataset and *(Bottom)* 3D dataset.

**Table 2:** Comparison on negative log-likelihood.

| Dataset | 2D dataset | | | 3D dataset | | |
|---|---|---|---|---|---|---|
| Training set | 1∼4 objects | 1&3 objects | | 1∼4 objects | 1&3 objects | |
| Test set | 1∼4 objects | 2 objects | 4 objects | 1∼4 objects | 2 objects | 4 objects |
| VAE | -13761.9 | -13801.4 | -13590.5 | -13712.0 | -13788.3 | -13433.2 |
| RICH | **-13890.3** | **-13908.3** | **-13796.6** | **-13818.0** | **-13867.3** | **-13539.7** |

### 4.3 OBJECT AND SCENE GENERATION

Apart from learning the part-level templates, RICH also has the ability to generate objects and scenes by recursively composing the learned templates. We show generation results in Figure 4. To generate the scenes, we first sample $\mathbf{z}_S^{\text{appr}} \sim \mathcal{N}(\mathbf{0}, \mathbf{I})$, and then sample other latents following the learned conditional prior distributions, and finally use the decoder to render the image. The objects are generated similarly, except that we decode up to the object level and ignore $\mathbf{z}_O^{\text{pres}}$ and $\mathbf{z}_O^{\text{pose}}$. We find that RICH has captured many predefined object types in the dataset, and also managed to come up with novel compositions. The generated scenes are also reasonable, with moderate distance and occlusion between objects.

### 4.4 GENERALIZATION PERFORMANCE

RICH represents the scene as composition of objects and parts. This naturally enables generalization to novel scenes. Here we evaluate RICH's capacity to generalize to scenes with novel number of objects. The training and validation sets of this task contain scenes of one and three objects only. We trained RICH and the VAE baseline again, and report the metrics in Table 1 and Table 2 on two test sets, one having two-object scenes only, and the other having four-object scenes only. We also show qualitative results in Figure 5 and Figure 6. As can be seen, RICH demonstrates quite decent generalization performance in both 2D and 3D scenes. We notice that in 3D case, there is a drop in recall when RICH is tested on four-object scenes. One reason is that four-object scenes exhibit more severe occlusion than the training set, and we have observed that when two objects are close and have occlusion, RICH would sometimes merge them into one object. Another reason is that in four-object scenes, objects are more likely to be partially outside the scene. In this case, RICH has difficulty predicting the precise object position, leading to unmatched predictions when we compute the recall.

### 4.5 DATA EFFICIENCY IN DOWNSTREAM TASKS

The compositional hierarchy learned by RICH can be useful in downstream tasks that require reasoning of part-object relationships. Here we consider a classification task. The input images are generated from the same distribution as described in Section 4.1 but with different random seeds. The label for each image is obtained by first counting the number of distinct parts within each object, and then summing the count over all objects. We expect the representation learned by RICH to

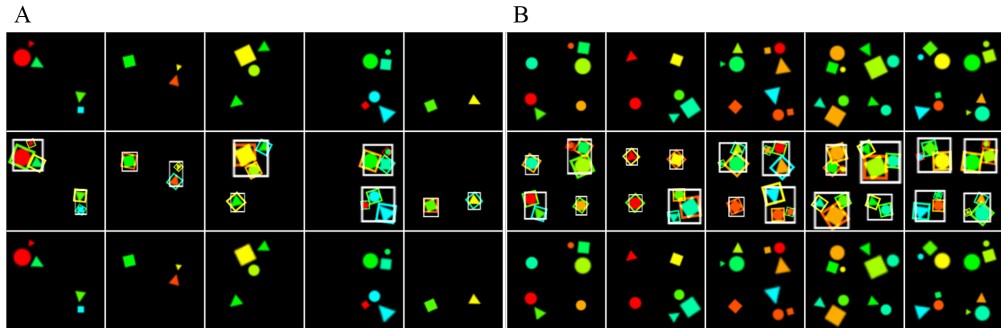

**Figure 5:** Generalization results on 2D dataset. RICH has been trained on scenes with 1 and 3 objects only, and tested on scenes with *(A)* 2 objects and *(B)* 4 objects. *(Top)* Input image. *(Middle)* Input image superimposed with predicted bounding boxes. *(Bottom)* Reconstruction.

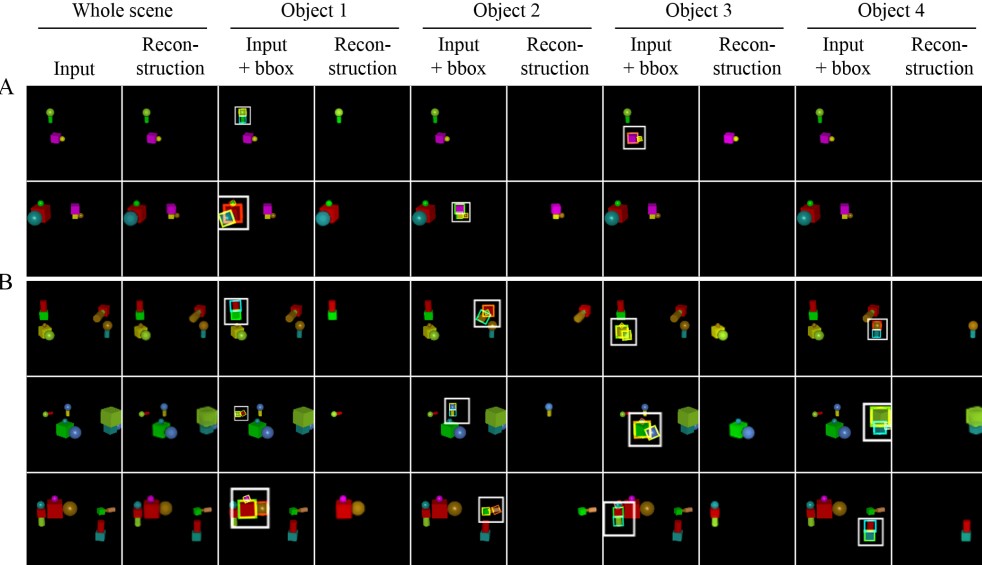

**Figure 6:** Generalization results on 3D dataset. RICH has been trained on scenes with 1 and 3 objects only, and tested on scenes with *(A)* 2 objects and *(B)* 4 objects. Each row shows the overall reconstruction, and the predicted bounding boxes and reconstruction from each object cell for a given input image.

bring better data efficiency compared to non-hierarchical baselines like SPAIR (Crawford & Pineau, 2019). To verify this, we use pretrained RICH and SPAIR to obtain the latent representation for each image, and then train a classifier on top of the representation to predict the label. Specifically, the unnormalized class probabilities $\mathbf{y}^{\text{RICH}}$ for RICH are computed as follows:

$$\mathbf{y}^{\text{RICH}} = \boldsymbol{f}_3^{\text{MLP}} \left( \sum_u \mathbf{w}_u \cdot \boldsymbol{f}_2^{\text{MLP}} \left( \sum_{v \in ch(u)} \mathbf{w}_v \cdot \boldsymbol{f}_1^{\text{MLP}} \left( \mathbf{z}_u^{\text{pres}} \cdot \mathbf{z}_v^{\text{pres}} \cdot \mathbf{M}[\mathbf{z}_v^{\text{addr}}] \right) \right) \right), \qquad (10)$$

where $u$ and $v$ are indices of objects and parts respectively, $\mathbf{w}_u$ and $\mathbf{w}_v$ are attention weights also computed by MLPs. Because RICH provides the compositional hierarchy, the classifier can explicitly merge information from parts that belong to the same object. For SPAIR that detects parts with no notion of objects, we include pose vectors as additional input for fair comparison. This allows the classifier to group parts into objects based on their positions. The unnormalized class probabilities $\mathbf{y}^{\text{SPAIR}}$ for SPAIR are computed as follows:

$$\mathbf{y}^{\text{SPAIR}} = \boldsymbol{f}_3^{\text{MLP}} \left( \sum_u \mathbf{w}_u \cdot \boldsymbol{f}_2^{\text{MLP}} \left( \sum_v \mathbf{w}_v^{(u)} \cdot \boldsymbol{f}_1^{\text{MLP}} \left( \mathbf{z}_v^{\text{pres}} \cdot \texttt{concat}[\mathbf{z}_v^{\text{appr}}, \mathbf{z}_v^{\text{pose}}] \right) \right) \right). \qquad (11)$$

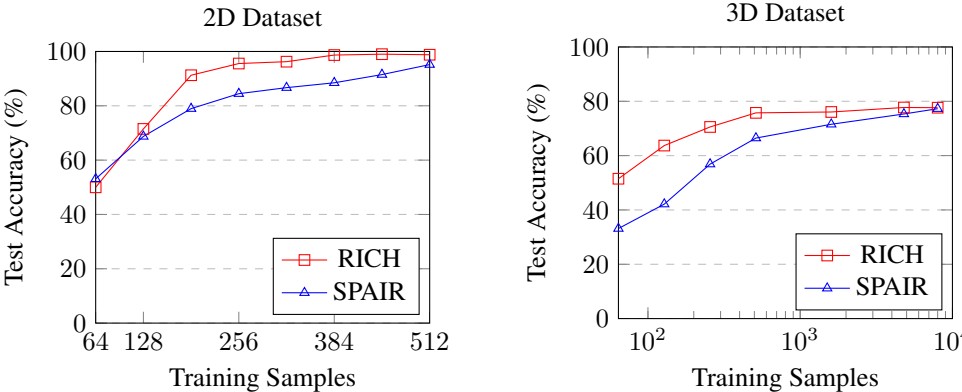

**Figure 7:** Comparison of data efficiency in downstream tasks.

The difference here is that we have multiple sets of attention weights $\mathbf{w}^{(u)}$, each should learn to select the parts that belong to one specific object.

We train the two classifiers using varying number of training samples. We ensure that the classifiers have comparable number of parameters. We use a parallel implementation of SPAIR that, as suggested by Anonymous (2020), has similar detection performance but faster training speed. We choose the best learning rate for both classifiers using a fixed validation set of size 12800. We report classification accuracy on a fixed test set also of size 12800. As shown in Figure 7, RICH representation approximately doubles the data efficiency on this downstream task compared to SPAIR representation.

## 5 CONCLUSION

We have proposed RICH, the first hierarchical generative model for learning interpretable compositional structures. RICH takes a top-down approach to infer the probabilistic scene graph representation for a given scene. This utilizes the higher-level appearance information to guide lower-level decomposition, thus avoiding the difficult routing problem faced by bottom-up approaches. Through extensive experiments, we have demonstrated that RICH is able to learn the compositional hierarchy from images containing multiple objects. An interesting future direction is to extend RICH to the sequential setting.

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

## A    DEALING WITH DEFORMABLE PRIMITIVES

In Section 4, we have demonstrated the effectiveness of RICH on two datasets of 2D and 3D scenes. However, these datasets only contain non-deformable primitives, meaning each primitive has only one possible shape. Here, we introduce a compositional MNIST dataset where the primitives are handwritten digits with considerable shape variation. Through the same set of experiments as in Section 4, we show that with slight modification, RICH can successfully deal with such deformable primitives.

### A.1    COMPOSITIONAL MNIST DATASET

We define the primitive parts to be individual digits from the MNIST database (LeCun et al., 1998). We have excluded the digit 1, since it can be used as sub-parts in other digits like 4 and 7, and thus may interfere with our definition of primitives. We then define nineteen types of objects, of which nine contain a single digit, six contain two digits, and the remaining four contain three digits. Here, the object definition specifies the class of each digit that an object should contain, and the relative position and orientation of these digits. We allow overlap among digits that constitute one object.

The dataset contains $128\times128$ color images, split into 64000 for training, 12800 for validation, and 12800 for testing. Each image describes a scene that contains one to four objects. To instantiate an object from its definition, we first sample the digits from the corresponding classes and resize them to $36\times36$. Next, we assign a random color to each digit. The colors are guaranteed to be distinct. Finally, we apply random scaling (within 10% of object size) and random rotation (within $\pm30$ degrees) to the object as a whole, and draw it at a random position in the scene. We ensure that different objects have minimal overlap, and that disjoint sets of digit samples are used for the training, validation, and test sets.

### A.2    MODIFICATIONS TO RICH

RICH decomposes the appearance vector of primitive $v$ into $\mathbf{z}_v^{\text{appr}} = (\mathbf{z}_v^{\text{addr}}, \mathbf{z}_v^{\text{what}})$, and decodes the mask $\hat{\mathbf{m}}_v$ and the image patch $\hat{\mathbf{x}}_v$ as $\hat{\mathbf{m}}_v = g(\mathbf{M}[\mathbf{z}_v^{\text{addr}}])$ and $\hat{\mathbf{x}}_v = \hat{\mathbf{m}}_v \odot h(\mathbf{z}_v^{\text{what}})$. Although in principle, such element-wise multiplication has the ability to model deformation, we empirically find this to be hard. In fact, $h(\mathbf{z}_v^{\text{what}})$ tends to produce an image patch filled with a single color, and is thus unable to distort the shape of template given by $\hat{\mathbf{m}}_v$.

To deal with deformable primitives, we add more inductive bias to the decoder. In particular, $\mathbf{z}_v^{\text{what}}$ is further split into $z_v^{\text{shape}}$ and $\mathbf{z}_v^{\text{color}}$ during the decoding process. $z_v^{\text{shape}}$ is designated to explain the shape variation, and is concatenated with the template embedding $\mathbf{M}[\mathbf{z}_v^{\text{addr}}]$ to modify the shape of template as $\hat{\mathbf{m}}_v = g(\texttt{concat}[\mathbf{M}[\mathbf{z}_v^{\text{addr}}], z_v^{\text{shape}}])$. As mentioned in Section 3.3, $\hat{\mathbf{m}}_v$ is in gray scale, so we then use $\mathbf{z}_v^{\text{color}}$ to colorize the modified template $\hat{\mathbf{m}}_v$ and produce $\hat{\mathbf{x}}_v = \hat{\mathbf{m}}_v \odot h(\mathbf{z}_v^{\text{color}})$.

One challenge here is to disentangle deformation from pose changes. Ideally, $z_v^{\text{shape}}$ should capture only the shape variation of primitive $v$ in its canonical pose, so that $z_v^{\text{shape}}$ and $\mathbf{z}_v^{\text{pose}}$ do not interfere with each other. This can be enforced by limiting the expressiveness of $z_v^{\text{shape}}$. In experiments, we find that restricting $z_v^{\text{shape}}$ to a one-dimensional scalar works reasonably well.

Another challenge arises during training. We have observed that RICH tends to split the digits into sub-parts at an early stage. Consequently, the strokes instead of the digits are learned as primitives, and the objects become composition of strokes, leading to an incorrect compositional hierarchy. To alleviate this problem, we take inspiration from SPACE (Anonymous, 2020) and introduce a color-filtered boundary loss for each glimpse $\mathbf{x}_v$, where $v$ is a primitive. The key observation is that if $\mathbf{x}_v$ captures only part of a digit, then there must be some pixels on the boundary of $\mathbf{x}_v$ that have the same color as the digit. Here, the thickness of boundary is a hyperparameter. We use $h(\mathbf{z}_v^{\text{color}})$ as a color filter to find such pixels. The boundary loss then penalizes the total number of those pixels. This drives $\mathbf{z}_v^{\text{pose}}$ to produce a larger glimpse region that reduces splitting. We anneal the weight of the boundary loss as training proceeds.

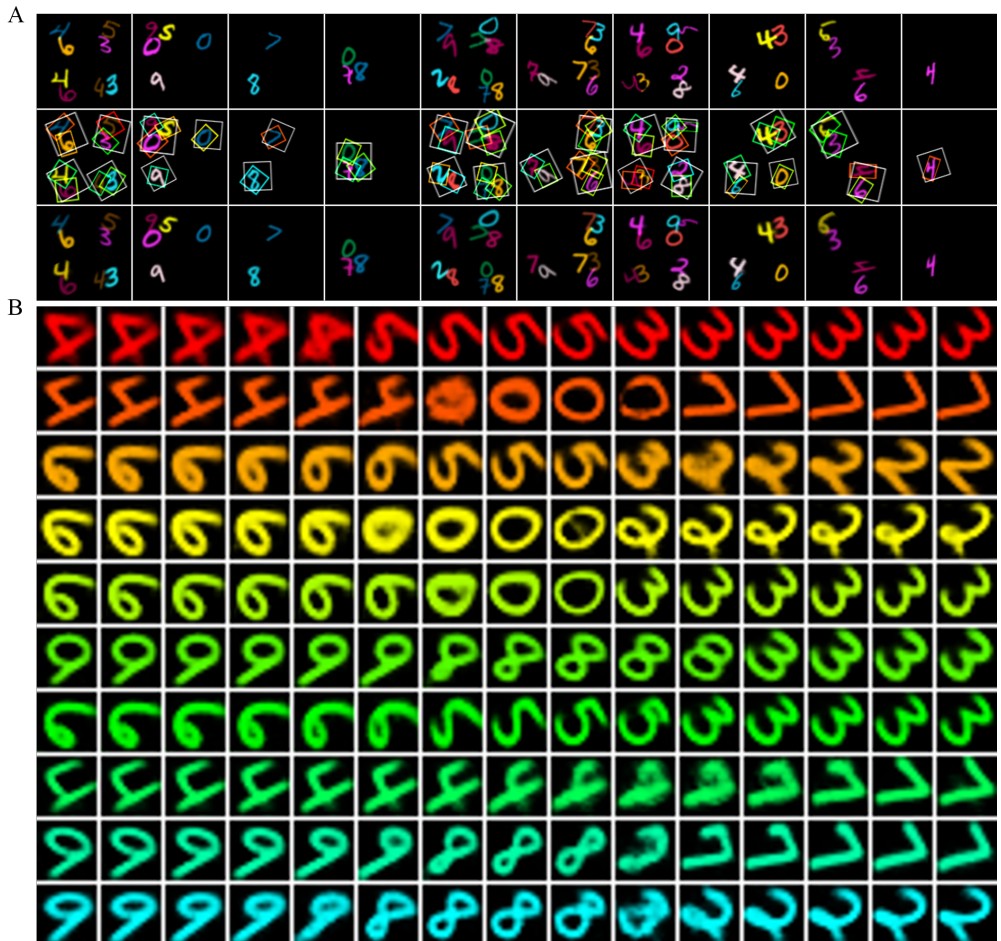

**Figure 8:** Qualitative results on compositional MNIST dataset. *(A) (Top)* Input image. *(Middle)* Input image superimposed with predicted bounding boxes. *(Bottom)* Reconstruction. *(B)* Learned part-level templates. Each row is decoded from the same template embedding with distortion specified by $z^{shape}$. Here for visualization purposes, we choose $z^{shape}$ values that are equally spaced within one standard deviation from the mean. Template colors indicate identity. Part bounding box colors indicate the chosen template.

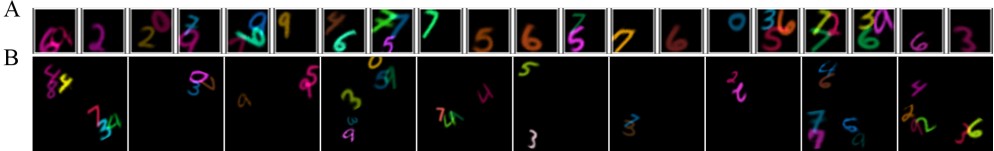

**Figure 9:** *(A)* Generated objects on compositional MNIST dataset. *(B)* Generated scenes on compositional MNIST dataset.

### A.3 EXPERIMENTAL RESULTS

Similar to Section 4, here we show both qualitative and quantitative results on the compositional MNIST dataset. Figure 8A shows qualitative results of scene decomposition. Again, the bounding boxes are fairly tight, demonstrating that the nodes in the probabilistic scene graph representation have been correctly associated with objects and parts in the images. We also visualize the learned part-level templates in Figure 8B. Here each row shows the templates decoded from one slot of the memory with varying degree of distortion. As can be seen, each slot captures visually similar digits, and all nine digits are captured. It is worth noting that the main purpose of memory is not to perfectly separate the different classes of digits, but to obtain the canonical pose for the digits. This

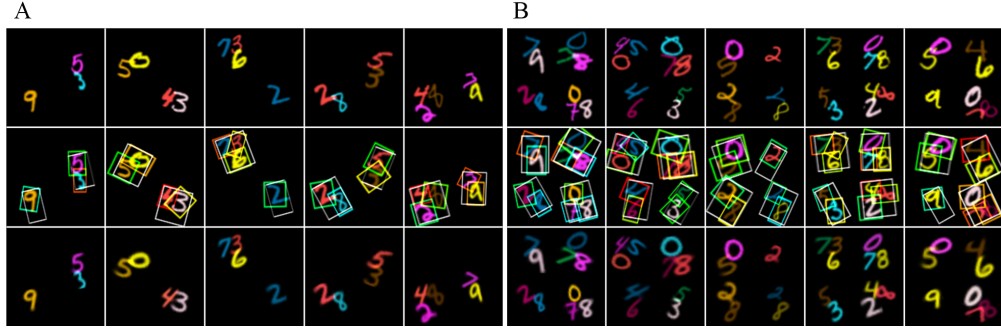

**Figure 10:** Generalization results on compositional MNIST dataset. RICH has been trained on scenes with 1 and 3 objects only, and tested on scenes with *(A)* 2 objects and *(B)* 4 objects. *(Top)* Input image. *(Middle)* Input image superimposed with predicted bounding boxes. *(Bottom)* Reconstruction.

**Table 3:** Quantitative results on object and part detection.

| Dataset | Compositional MNIST dataset | | |
|---|---|---|---|
| Training set | 1~4 objects | 1&3 objects | |
| Test set | 1~4 objects | 2 objects | 4 objects |
| Object count error | 0.0015 | 0.00064 | 0.00027 |
| Object precision | 0.9963 | 0.9899 | 0.9365 |
| Object recall | 0.9962 | 0.9898 | 0.9364 |
| Part count error | 0.016 | 0.0086 | 0.023 |
| Part precision | 0.9981 | 0.9989 | 0.9982 |
| Part recall | 0.9983 | 0.9990 | 0.9989 |

is essential for the scene graph representation to infer consistent orientation for the digits. Figure 8B shows that this purpose is indeed achieved, since the digits have almost the same pose within each memory slot.

Figure 9 shows the generated objects and scenes, which resemble the training images. Figure 10 shows qualitative generalization results. The learned compositional hierarchy generalizes well to unseen number of objects. We report quantitative results in Table 3 and Table 4. The counting error, precision, and recall are mainly to show that the representation is learned properly. We compare negative log-likelihood with a VAE baseline to demonstrate that the probabilistic scene graph representation can be obtained without impairing generation quality.

Finally, we consider a downstream task that requires reasoning of part-object relationships, and demonstrate the transferability of RICH representation. The task setup, model architecture, and training procedure are similar to those described in Section 4.5. The difference here is that we seek to predict the sum of max digit within each object, and we treat it as a 37-way classification task (possible output can be integers in [0, 36]). Experimental results in Figure 11 show that compared to SPAIR, the representation learned by RICH leads to approximately doubled data efficiency.

**Table 4:** Comparison on negative log-likelihood.

| Dataset | Compositional MNIST dataset | | |
|---|---|---|---|
| Training set | 1~4 objects | 1&3 objects | |
| Test set | 1~4 objects | 2 objects | 4 objects |
| VAE | -13096.4 | -12915.7 | -10832.9 |
| RICH | **-13377.1** | **-13436.9** | **-12801.6** |

**Figure 11:** Comparison of data efficiency in downstream task.

