# OpenReview forum: "Generative Hierarchical Models for Parts, Objects, and Scenes"
_ICLR.cc/2020/Conference — Reject_

### Official Review · AnonReviewer3 · 2019-10-22
**Official Blind Review #3**

**Rating:** 3

**Review:**

Contributions: this submission proposes a generative framework which employs a hierarchical decomposition of scene, objects and parts. It modifies the SPAIR framework (Crawford & Pineau) and replaces the recurrent generation with parallel generation, by assuming the number of objects is known and fixed. Results are presented on two synthetic datasets with 2D or 3D shapes.

Assessment:
- The proposed model and learning framework are closely related to previous work (AIR, SPAIR). The authors replaced the "sequential processing ... with single forward pass", which requires the number of objects to be fixed.
- In the proposed model, there is no constraint on the part-object hierarchy, and it is unclear whether it can learn to discover such hierarchies directly by reconstructing single static images. It is also not evaluated in the submission.
- There are no comparisons with AIR or SPAIR on the two datasets the authors created.
- The authors are recommended to compare with previous work on hierarchical generative modelings with objects and parts, such as (Xu et al, ICLR 2019), and other related work, such as SPIRAL (Ganin et al.)

Due to the limited contribution, lack of comparison with related work and limited empirical evaluation, I recommend rejection of the submission.

[1] Xu el al, Unsupervised Discovery of Parts, Structure, and Dynamics, ICLR 2019.
[2] Ganin et al., Synthesizing Programs for Images using Reinforced Adversarial Learning.


--------------------------------
Post-rebuttal:
Thank you for your detailed answers to my questions and updated manuscript. The writing has indeed significantly improved and some questions (e.g. varying number of objects) have been addressed. After reading the rebuttal, my concerns remain that: (1) lack of empirical evaluation on more complex real / synthetic datasets; (2) it is still unclear to me how such hierarchy is inferred from single images.

**Experience Assessment:**

I have published one or two papers in this area.

**Review Assessment: Checking Correctness Of Derivations And Theory:**

I assessed the sensibility of the derivations and theory.

**Review Assessment: Checking Correctness Of Experiments:**

I assessed the sensibility of the experiments.

**Review Assessment: Thoroughness In Paper Reading:**

I read the paper at least twice and used my best judgement in assessing the paper.

---

> ### Author Response · Authors · 2019-11-15
> **Response to Review #3**
>
> Thank you for the helpful comments. We will respond in order.
>
> 1) We believe the key to dealing with variable number of objects is the z_pres variable. It can turn on/off individual object representations. Although we use parallel inference, we still have the z_pres variable, so we do not require the number of objects to be fixed. Also, our datasets do contain variable number of objects. We have clarified this in the first paragraph of Section 3.3. We conjecture that our top-down inference approach enables the parallel inference to perform well, since higher-level appearance information can guide lower-level decomposition. Please also kindly refer to general response [A1] - [A2] for the main contribution and novelty of our paper.
>
> 2) We have updated the paper to clarify our representation of the compositional hierarchy. As shown in Figure 1A, the compositional hierarchy is represented as a latent tree. Both the tree structure and the latent variables are inferred for each static image in the dataset. One way to visualize the tree is to draw bounding boxes around objects and their constituent parts according to the inferred tree structure and pose vectors. Figure 1B illustrates this, and our qualitative results should be interpreted in the same way. Since we recursively apply spatial transformer during inference, we have the implicit constraint that parts should be within the object bounding box.
>
> 3) Regarding empirical evaluation and ablations / baselines, please kindly refer to general response [A5] - [A8].
>
> 4) Please kindly refer to general response [A10] - [A11] for a brief comparison.

---

### Official Review · AnonReviewer2 · 2019-10-23
**Official Blind Review #2**

**Rating:** 3

**Review:**

Update: I thank the auhors for the detailed response. The updated paper does look more convincing, but my main concern remains - all considered datasets and tasks are synthetic and specifically made for the method. I think experiments on more real data would be crucial to show the potential of the method. Some examples I can imagine would be part segmentation in images or 3D models, or robotic control.

---

The paper proposes an unsupervised approach to learning objects and their parts from images. The method is based on the "Attend, Infer, Repeat" (AIR) line of work and adds a new hierarchy level to the approach, corresponding to object parts. The method is evaluated on two custom synthetic dataset: one composed of simple 2D geometric shapes and another one with 3D geometric shapes. On these datasets the method successfully infers the object-part structure and can parse and reconstruct provided images.

While the general topic of the paper is interesting, I do not think it is fit for publication. First and foremost, the experiments are very incomplete: the method is only evaluated on two custom synthetic datasets and not compared against any baselines or ablated versions of the method. Moreover, the proposed approach seems like a relatively minor modification of SPAIR (Crawford & Pineau).

Pros:
1) The topic is interesting and the method generally makes sense.
2) The method works on the tasks studied in the paper, including relatively challenging scenarios with 3D occlusions or ambiguity in assigning parts to objects.
3) The paper shows generalization to a number of objects different from that seen during training.

Cons:
1) Experiments are limited:
1a) The method is evaluated on custom and relatively toy datasets. It is unclear if it would apply to more practical situations. While I agree that at the high level the question of decomposition of object into parts is interesting, it is still important to connect research in this direction to potential practical applications. Where could the proposed method be used? Perhaps in some control tasks, such as robotics? Could it be applied to more realistic data, such as for instance ShapeNet objects?
1b) There are no comparisons to baselines. One might argue that baselines do not exist since there are no methods addressing the same task. Generally, it is the job of the authors to come up with relevant baselines to show that the proposed model actually improves upon some simpler methods. One variant of obtaining baselines is via ablating the proposed model. Another is by taking prior approaches, for instance object-based ones without the part decomposition, and showing that the part decomposition allows to improve upon those in some sense.

2) The novelty is somewhat limited: the method seems like a relatively straightforward extension of SPAIR by adding another hierarchy layer. It might be sufficient if the experimental results would be strong, but given that they are not, becomes somewhat concerning. If the method does indeed include significant technical innovation, it might be helpful to better highlight it.

3) The related work overview is extremly limited. It is authors' job to provide a comprehensive ovreview of prior literature and I cannot do it for them here, but below are a few papers that come to mind. This is by no means a complete list.

[1] Zhenjia Xu, Zhijian Liu, Chen Sun, Kevin Murphy, William T. Freeman, Joshua B. Tenenbaum, Jiajun Wu. Unsupervised Discovery of Parts, Structure, and Dynamics. ICLR 2019
[2] Shubham Tulsiani, Hao Su, Leonidas J. Guibas, Alexei A. Efros, Jitendra Malik. Learning Shape Abstractions by Assembling Volumetric Primitives. CVPR 2017
[3] Jun Li, Kai Xu, Siddhartha Chaudhuri, Ersin Yumer, Hao Zhang, Leonidas Guibas. GRASS: Generative Recursive Autoencoders for Shape Structures. SIGGRAPH 2017.
[4] Gopal Sharma, Rishabh Goyal, Difan Liu, Evangelos Kalogerakis, Subhransu Maji. CSGNet: Neural Shape Parser for Constructive Solid Geometry. CVPR 2018.
[5] Adam R. Kosiorek, Hyunjik Kim, Ingmar Posner, Yee Whye Teh. Sequential Attend, Infer, Repeat: Generative Modelling of Moving Objects. NeurIPS 2018.

4) Presentation is at times suboptimal:
4a) It would be very helpful to have more visuals and intuitions about the functioning of the method, as opposed to equations. Equations are definitely good to have, but they are not the easiest to parse, especially by those not intimately familiar with this specific line of work.
4b) It is quite unclear to me how and why is the memory used in the model. If this is described in another paper, it would be useful to point there, but still briefly summarize in this paper to make it self-contained.

**Experience Assessment:**

I have published one or two papers in this area.

**Review Assessment: Checking Correctness Of Derivations And Theory:**

I assessed the sensibility of the derivations and theory.

**Review Assessment: Checking Correctness Of Experiments:**

I carefully checked the experiments.

**Review Assessment: Thoroughness In Paper Reading:**

I read the paper at least twice and used my best judgement in assessing the paper.

---

> ### Author Response · Authors · 2019-11-15
> **Response to Review #2**
>
> Thank you for your constructive review and for taking the time to provide a reference list. We will respond in order.
>
> 1a) We agree that it is important to connect to practical applications. However, this line of research has not been close to the level of applying to robotics or real-world datasets. This is due to the challenging problem setting: (i) We use generative modeling, which learns both representation and rendering process; (ii) Our latent representation is hierarchical, compositional, and interpretable at the same time, which no prior work has achieved; (iii) Our model is end-to-end trainable via purely unsupervised learning. In our revised paper, we have introduced a compositional MNIST dataset that presents an additional challenge of shape variation compared to our 2D and 3D datasets. We think this is one characteristic of more realistic data, and we have shown promising results.
>
> 1b) Regarding empirical evaluation and ablations / baselines, please kindly refer to general response [A5] - [A8].
>
> 2) Thanks for the suggestion. We have revised our paper to better highlight the significance. Please also refer to general response [A1] - [A2] for a summary of the contribution and novelty of our paper.
>
> 3) We totally agree and have included more discussion on related work. Please kindly refer to general response [A10] - [A13] for a brief comparison.
>
> 4a) Thanks for the suggestion. We have updated the notation and equations to make them more accessible. We have also added more intuitive explanation and illustrations to clarify the formulation and contribution of our model.
>
> 4b) We have included a dedicated paragraph in Section 3.3 to describe the memory. In short, memory is able to capture the canonical pose, which is essential to the scene graph that describes composition based on pose transformations. The learned memory templates also bring additional interpretability to our model.

---

### Official Review · AnonReviewer4 · 2019-10-27
**Official Blind Review #4**

**Rating:** 3

**Review:**

-- Update --
Thank you for the detailed response and updates. The quality has improved but as mentioned in the original review, it would be good to see results on a more significant non-synthetic task before I would argue for acceptance.

------------------
The authors propose a generative model with a hierarchy of latent variables corresponding to a scene, objects, and object parts. The method is evaluated on a synthetic dataset with manual inspection, MSE, and object counting performance. The method can form bounding boxes around objects and parts in the examples that are shown, and the MSE is similar to that of a VAE.

The idea of decomposing a probabilistic model hierarchically is potentially interesting, but this paper has drawbacks in terms of experimental quality, significance, and presentation.

-- Experiments --
The experiments are done on a manually constructed synthetic dataset, so it is not clear whether the proposed method would work in more realistic or more challenging settings. For instance, the dataset was constructed using a recursive process which probably does not resemble a realistic distribution of objects and their parts, and the parts are well-separated within the object.

The experiments on the synthetic dataset are not thorough, and could be more interesting with ablations, evaluating performance as the dataset parameters vary (e.g. number of objects, amount of occlusion), better baselines, and an evaluation metric rather than manual inspection.

-- Significance --
With respect to significance, the formulation is fairly straightforward and builds off of previous techniques introduced in e.g. SPAIR. The lack of demonstrating usefulness on a downstream application limits this paper's significance; for instance the SPAIR paper used the generative model as a front-end for a card game and Atari game. The paper should more clearly demonstrate the benefits/tradeoffs of the hierarchical aspect.

-- Presentation --
The paper leaves many terms undefined so the paper is not self contained, e.g. the 'problem of routing', 'memory buffer', 'memory template', 'variational memory addressing', 'spatial transformer' (missing a reference here). I suspect the related work is missing references, e.g. "Unsupervised Discovery of Parts, Structure, and Dynamics" Xu et al ICLR 2019. Finally, the paper should be proof-read for grammatical errors.


**Experience Assessment:**

I have published one or two papers in this area.

**Review Assessment: Checking Correctness Of Derivations And Theory:**

I assessed the sensibility of the derivations and theory.

**Review Assessment: Checking Correctness Of Experiments:**

I assessed the sensibility of the experiments.

**Review Assessment: Thoroughness In Paper Reading:**

I read the paper at least twice and used my best judgement in assessing the paper.

---

> ### Author Response · Authors · 2019-11-15
> **Response to Review #4**
>
> Thank you for the constructive review. We will respond in order.
>
> -- Experiments --
> Our problem setting is already challenging: (i) We use generative modeling, which learns both representation and rendering process; (ii) Our latent representation is hierarchical, compositional, and interpretable at the same time, which no prior work has achieved; (iii) Our model is end-to-end trainable via purely unsupervised learning. We believe that research under such problem setting has not been close to the level of applying to real-world datasets. We also would like to note that our datasets, though manually constructed, present challenges of multi-scale and multi-pose entities with occlusion and compositional ambiguity. Specifically, in our 3D dataset, there is occlusion between parts, so parts are not well-separated. In our revised paper, we have shown promising results on a compositional MNIST dataset that presents an additional challenge of shape variation.
>
> Regarding empirical evaluation and ablations / baselines, please kindly refer to general response [A5] - [A8]. We have actually provided NLL, counting error, precision, and recall as evaluation metrics. We appreciate your suggestion to evaluate performance as the dataset parameters vary. In the generalization experiments, we have already tested performance on different number of objects.
>
> -- Significance --
> Thanks for your suggestion. We have included experiments on downstream tasks in our revised paper. We show that compared to SPAIR, our proposed hierarchical representation approximately doubles the data efficiency on these downstream tasks. Please also refer to general response [A1] - [A2] for the contribution and novelty of our paper.
>
> -- Presentation --
> Thanks for the feedback. We have improved our presentation quality and included more discussion on related work. Please kindly refer to general response [A10] for a brief comparison with Xu et al.

---

### Author Response · Authors · 2019-11-15
**General Response to All Reviewers (Part 1 / 3)**

We would like to thank all reviewers for their helpful comments and for pointing to relevant literature. Here we address the main concerns shared by all reviewers.

1) Limited significance: the proposed method just adds another layer to SPAIR and lacks novelty; hence, experiments on more realistic data / downstream tasks are needed to demonstrate significance.

[A1] We have uploaded a revised version of our paper, where we highlight our contributions. We aim to learn interpretable representation for compositional hierarchies. To this end, we propose a probabilistic scene graph representation that describes the compositional hierarchy as a latent tree. The nodes in the tree represent the appearance of entities, and the edges specify the relative pose at each level of composition. To ensure proper learning, we design a decoder for the probabilistic scene graph that closely follows the rendering process in computer graphics. We also present a top-down inference approach that is able to simultaneously infer both the tree structure and the latent appearance and pose vectors.

[A2] We believe that our generative modeling of the compositional hierarchy is novel, and that obtaining such interpretable representation from raw pixels through end-to-end purely unsupervised learning is highly nontrivial. Also, our model is fully general and does not have to be limited to three levels. We do not think simple extensions of SPAIR could achieve what we have done, nor are we aware of any prior work that is able to learn hierarchical, compositional, and interpretable latent representation at the same time. Therefore, we believe that our design of the probabilistic scene graph representation and our proposed inference and learning methods constitute significant contributions.

[A3] We agree that experiments on more realistic data will add to the significance of our paper. Since our 2D and 3D shape datasets already contain multi-scale and multi-pose entities with occlusion and compositional ambiguity, we think one way to make a more realistic dataset is to include deformable primitives. Hence, in the appendix of our revised paper, we introduce a compositional MNIST dataset where the primitives are handwritten digits with considerable shape variation. We show that with slight modification, our model can successfully deal with such deformable primitives.

[A4] We also agree that the benefit of our proposed hierarchical representation would be better demonstrated on downstream tasks. We have included two such tasks in our revised paper. For the 2D and 3D shape datasets, we make a label for each image, and train a classifier on top of the learned representation to predict the label. The label is obtained by first counting the number of distinct parts within each object, and then summing the count over all objects. Similarly, for the compositional MNIST dataset, we seek to predict the sum of max digit within each object. We use SPAIR as a non-hierarchical baseline, and tune it to detect parts. Experimental results show that compared to SPAIR, our proposed hierarchical representation approximately doubles the data efficiency on these downstream tasks.

---

### Author Response · Authors · 2019-11-15
**General Response to All Reviewers (Part 2 / 3)**

2) Insufficient empirical evaluation: ablations / better baselines are needed.

[A5] As we have explained in [A2], our main contribution is the design and successful learning of a probabilistic scene graph representation, which no prior work has accomplished. The main purpose of learning such representation is not to improve detection performance. Instead, the representation itself is of significance, because it can bring better compositionality, interpretability, transferability, and generalization ability. Therefore, in the experiments, we have focused on showing that the representation can be learned properly by our proposed methods, and that the learned representation does bring some of the benefits. We do not agree that lack of comparison with baselines on detection performance can be grounds for insufficient empirical evaluation.

[A6] To be more specific, the counting error, precision, and recall are mainly to show that the representation is learned properly, i.e., the nodes in the latent tree associate correctly with the parts and objects. Together with the qualitative results, it can be seen that the compositional hierarchy is well captured by our model, even in the presence of occlusion and compositional ambiguity. In terms of compositionality and interpretability, we are not aware of any baseline that is at the same level as our model. Taking SPAIR as an example, it is clear that SPAIR cannot simultaneously detect parts and objects, let alone model the part-object composition. Besides, SPAIR lacks the discrete interpretable representation that our model provides to capture the identity and canonical pose for individual parts. In terms of transferability, our new experiments on downstream tasks show that the proposed model improves data efficiency over SPAIR.

[A7] Regarding the negative log-likelihood (NLL), we include it mainly to show that while the probabilistic scene graph representation brings so many benefits as mentioned above, it can be obtained without impairing NLL or generation quality. This is not trivial as it may appear, because compositionality and interpretability introduce constraints on the model structure and discreteness in the latent variables. Hence, we are faced with a more difficult optimization problem in a more limited model space, compared to non-hierarchical, continuous representations. Therefore, we do not consider improving NLL as our main goal, though our model seems to have achieved better NLL as a side effect. We also believe that AIR and SPAIR would have worse generation quality, because they need to set a very low success probability for the prior of z_pres to encourage sparsity. This can often lead to the generated image being all black, i.e., containing no object. In contrast, our model learns this prior conditioning on higher-level z_appr, and thus avoid this problem.

[A8] Similar arguments apply to ablations. Our main contribution is not to improve detection performance or NLL, but to obtain the probabilistic scene graph representation. Hence, we feel it unnecessary to compare detection performance or NLL with ablations that break the scene graph representation. In this sense, the only ablation that may seem reasonable is to remove the memory that learns primitive templates. However, we believe that memory is not redundant. The scene graph relies on the relative pose between entities to describe composition. To obtain the relative pose, the model needs representation of the canonical pose. This is achieved by the memory, as we have shown in the qualitative results. Without memory, there will be no notion of canonical pose, leaving the scene graph not well defined.

---

### Author Response · Authors · 2019-11-15
**General Response to All Reviewers (Part 3 / 3)**

3) Incomplete overview of related work.

[A9] We agree that we need to discuss more related work, and we thank reviewers for taking the time to list relevant references. Here we briefly compare our model with those pointed by reviewers.

[A10] Xu et al [1] focus on motion decomposition from image pairs. They obtain the part hierarchy based on consistency of motion. The latent space of their model contains only the representation for motion. The part appearance and the hierarchical structure are not modeled as latent variables. Also, this latent space does not seem interpretable. In contrast, we model both the hierarchical structure and the object pose and appearance as latent variables. This richer latent representation is learned from single images, i.e., by merely observing the possible static compositions with no motion information.

[A11] Sharma et al [2] and Ganin et al [3] learn to parse an image into a program that, when fed to a rendering engine, can generate the image. Their models lack the explicit representation of compositional relationships among primitives. Also, the primitives are pre-defined by the rendering engine. In contrast, our model automatically discovers the primitives and can learn the renderer.

[A12] Tulsiani et al [4] and Li et al [5] operate directly in 3D space, which is quite a different problem setting. We appreciate the symmetry modeling in [5], which is complementary to our model. However, we note that neither of these two models has a hierarchically structured latent space.

[A13] Kosiorek et al [6] extend AIR to model image sequences. We find the approach inspiring, and also consider extending our model to the sequential setting as an interesting future direction.

4) Suboptimal presentation quality.

[A14] We have greatly improved our writing. Specifically, we have updated the notation and equations to make them more accessible. We have added more intuitive explanation and illustrations to clarify the formulation and contribution of our model. To make our paper more self-contained, we have also included a paragraph that describes the memory. We sincerely hope that reviewers could take the time to read our paper again, and we would appreciate any additional feedback.


[1] Zhenjia Xu, Zhijian Liu, Chen Sun, Kevin Murphy, William T. Freeman, Joshua B. Tenenbaum, Jiajun Wu. Unsupervised Discovery of Parts, Structure, and Dynamics. ICLR 2019.
[2] Gopal Sharma, Rishabh Goyal, Difan Liu, Evangelos Kalogerakis, Subhransu Maji. CSGNet: Neural Shape Parser for Constructive Solid Geometry. CVPR 2018.
[3] Yaroslav Ganin, Tejas Kulkarni, Igor Babuschkin, S.M. Ali Eslami, Oriol Vinyals. Synthesizing Programs for Images using Reinforced Adversarial Learning.
[4] Shubham Tulsiani, Hao Su, Leonidas J. Guibas, Alexei A. Efros, Jitendra Malik. Learning Shape Abstractions by Assembling Volumetric Primitives. CVPR 2017.
[5] Jun Li, Kai Xu, Siddhartha Chaudhuri, Ersin Yumer, Hao Zhang, Leonidas Guibas. GRASS: Generative Recursive Autoencoders for Shape Structures. SIGGRAPH 2017.
[6] Adam R. Kosiorek, Hyunjik Kim, Ingmar Posner, Yee Whye Teh. Sequential Attend, Infer, Repeat: Generative Modelling of Moving Objects. NeurIPS 2018.

---

### Decision · Program_Chairs · 2019-12-19

**Decision:**

Reject

**Comment:**

The authors proposes a generative model with a hierarchy of latent variables corresponding to a scene, objects, and object parts.

The submission initially received low scores with 2 rejects and 1 weak reject.  After the rebuttal, the paper was revised and improved, with significant portions of the paper completely rewritten (the description of the model was rewritten and a new experiment comparing the proposed model to SPAIR was added).  While the reviewers acknowledged the improvement in the paper and accordingly adjusted their score upward, the paper is still not sufficiently strong enough to be accepted (it currently has 3 weak rejects).

The reviewer expressed the following concerns:
1. The experiments uses only a toy dataset that does not convincingly demonstrate the generalizability of the method to more realistic/varied scenarios. In particular, the reviewers voiced concern that the dataset is tailored to the proposed method

2. Lack of comparisons with baseline methods such as AIR/SPAIR and other work on hierarchical generative models such as SPIRAL.
In the revision, the author added an experiment comparing to SPAIR, so this is partially addressed.  As a whole, the paper is still weak in experimental rigor.  The authors argue that as their main contribution is the design and successful learning of a probabilistic scene graph representation, there is no need for ablation studies or to compare against baselines because their method "can bring better compositionality, interpretability, transferability, and generalization".  This argument is unconvincing as in a scientific endeavor, the validity of such claims needs to be shown via empirical comparisons with prior work and ablation studies.

3. Limited novelty
The method is a fairly straightforward extension of SPAIR with another hierarchy layer. This would not be a concern if the experimental aspects of the work was stronger.

The AC agrees with the issues pointed by the reviewers.  In addition, the initial presentation of the paper was very poor.  While the paper has been improved, the changes are substantial (with the description of the method and intro almost entirely rewritten). Regardless, despite the improvements in writing, the paper is still not strong enough to be accepted.  I would recommend the authors improve the evaluation and resubmit.